# Sport for Development Programs Contributing to Sustainable Development Goal 5: A Review

Yong-Yee Chong [1], Emma Sherry [2], Sophia Harith [1,3] and Selina Khoo [1,*]

1    Centre for Sport and Exercise Sciences, Universiti Malaya, Kuala Lumpur 50603, Malaysia;
     yongyeechong@um.edu.my (Y.-Y.C.); sophia.harith@lunex-university.net (S.H.)
2    Department of Management and Marketing, Swinburne University of Technology,
     Hawthorn, VIC 3122, Australia; esherry@swin.edu.au
3    Department of International Sport Management, LUNEX International
     University of Health, Exercise, and Sports, 4671 Differdange, Luxembourg
*    Correspondence: selina@um.edu.my

**Abstract:** In Sport for Development (SFD), sport is used as a cost-effective tool to facilitate the objectives of various organizations, not limited to increasing access to education, youth development, social cohesion, and gender equality. This review aims to systematically analyze SFD programs that contribute to gender equality and women empowerment under Sustainable Development Goal 5 (SDG 5). The PRISMA methodology was used to guide the screening and selection process. Fifteen studies were identified from the Web of Science, Scopus, and SPORTDiscus databases, the Journal of Sport for Development, forward–backward reference searches, and manual searches on four prominent sport, gender, and development researchers. The findings indicated that there was evidence of micro-level outcomes in every study and three achieved meso-level impact; however, none of these studies' suggested changes have reached the macro-level of impact when the outcomes were reported in these articles. There was a lack of intervention studies that investigated the mechanisms and reported outcomes through a validated monitoring and evaluation process. This review provides significant insights into: (a) identifying future SFD research areas, (b) refining SFD program evaluations, (c) developing indicators of outcomes for sport programs contributing to SDG 5, and (d) reproducing sustainable development outcomes under SDG 5.

**Keywords:** sustainable development; SGD 5; gender equality; women empowerment; sport for development; monitoring and evaluation



## 1. Introduction

Sport for Development (SFD) refers to sport-based interventions designed to contribute towards non-sport goals [1]. These non-sport goals are social development goals, related to education, health, gender, livelihoods, disability, peace, and social cohesion [2]. SFD practitioners aspire to disrupt the status quo of the existing social systems, where inequity is often found to contribute to social development challenges [3]. In this review, we focus on the key components of Sustainable Development Goal 5 (SDG 5): gender equality and women empowerment. Empowerment is achieved when one is able to make strategic life choices after being denied the ability to do so [4]. In the context of gender, inequality is the difference felt by two persons/groups of different genders; inequity refers to unfair, problematic treatment caused by injustice against a gender [5,6]. In SFD, leveling the playing field for women through sports has been well received but the progress towards gender equality has been deemed to be slow [7], especially for structural changes that are only attainable with macro-level intervention outcomes. Having said that, to enact structural change, the complexities at the community and societal levels have to be considered and solutions have to be directed at all levels [8]. At the individual level, the "Girl Effect" is particularly apparent in many girls empowerment interventions [9–11], where girls are

expected to play the sole and progressive role towards gender equality, neglecting the complex local context that could limit the sustainable changes in their community [12–14]. The empowerment of individuals is also attainable beyond the intervention program. In the context of research design, the agency of women and girls can be further exercised through their inclusion in the design and evaluation of the program, being more than the participants of sport activities [15]. Coalter [16] argued that a 'displacement of scope' happens in the SFD research, by which evaluators equate micro-level impacts such as sport participation with macro-level or societal change. As such, SFD researchers are urged to increase the rigorousness and clarity of these studies, especially when reporting outcomes beyond the "sport" touchline [17–19]. As the field of SFD grows substantially, LeCrom and Martin [20] argue that scholars should focus on process-based research on the development and management of SFD instead of solely on program evaluations. Our review investigated how SFD outcomes are presented in the literature and the role of evaluation in these gender equality and women empowerment programs since the global SFD movement was set in motion at the turn of the 21st century.

### 1.1. Research Aim

Although the SFD movement and associated research has grown substantially over the past two decades, there remains a paucity of empirical evidence about the contribution of sport in gender-based development [2]. De Soysa and Zipp [7] argued it is timely to produce a more systematic and macro-level understanding of the field, which can help to elucidate gaps and determine the direction of future research. This paper aims to systematically review the literature that investigates the contribution of sport programs towards gender equality and women empowerment. Through the findings, we provide significant insights into the programing and evaluation of outcomes under SDG 5.

First, our foci are situated on studies that investigate the characteristics of SFD programs for gender equality and women empowerment, and how the outcomes of these programs were synthesized and presented. Second, we look at the evidence of these program outcomes to understand if they were measured at the individual (micro-), community (meso-), or societal (macro-) level. Based on Lyras and Welty Peachey [21], the intended social change through sport-based interventions are achievable across macro- and micro-levels; as such, we intend to describe the multiple levels of program outcomes interpreted as gender equality and women empowerment within the purview of SDG 5. The United Nations Educational, Scientific, and Cultural Organization expanded on the empowerment of women, stating that overcoming three levels of barriers is necessary for empowerment to materialize. These encompass the micro-level (personal or psychosocial), meso-level (professional and institutional), and macro-level (policy and strategy) [22]. The empowerment of the gender that suffers from injustice is fundamental towards gender equality [23]. Gender is a multifaceted variable affected by social, cultural, and economic factors on the same platform where men and women co-exist [24,25]. For this review, we followed the indicators and guidelines proliferated by the Commonwealth Secretariat [26] and the United Nations Educational, Scientific, and Cultural Organization [22] to interpret the outcomes of the programs.

### 1.2. The Outcome Indicators in Gender Equality and Women Empowerment

Kim [27] posited that the advancement of gender equality in certain countries is slow; some countries have experienced a widening gender gap despite long-term development efforts. De Soysa and Zipp [7] argued that the gender equality movement within sports then converged with the emergence of SFD, at the juncture when the Millennium Development Goals (MDGs) were formalized by the United Nations in 2000. In the following years, the United Nations strategically launched the United Nations Office for Sport for Development in 2001 and declared 2005 as the International Year of Sport and Physical Education. Notably, promoting gender equality and empowering women was introduced as one of

the eight MDGs [28], succeeded by the Sustainable Development Goals (SDGs) in the post-2015 agenda.

Nine targets and 14 indicators were developed to measure the progress towards SDG 5—to achieve gender equality and empower all women and girls. Initially, the associated targets and indicators made no explicit mention of sport [29,30]. Thereafter, the Commonwealth Secretariat's analysis on the contribution of SFD towards the sustainable development agenda identified four targets where sport can be utilized as a development tool [26]. The highlighted targets are eliminating all forms of discrimination against women and girls (Target 5.1), eliminating all forms of violence against women and girls (Target 5.2), women achieving full and effective participation and equal opportunities for leaderships at all levels of decision making (Target 5.5), and implementing policies and enforceable legislation for the promotion of gender equality and the empowerment of all women and girls (Target 5c).

In 2017, the United Nations Educational, Scientific, and Cultural Organization adopted the Kazan Action Plan to stress the commitment to link sport policy development to the 2030 agenda of the United Nations. Among the key actions were to "conduct a feasibility study on the establishment of Global Observatory for Women, Sport, Physical Education, and Physical Activity" as well as to "develop common indicators for measuring the contribution of physical education, physical activity, and sport to prioritized SDGs and their targets" [31] (p. 5). The importance of gender equality and women empowerment were underlined:

"Gender equality and empowerment of women and girls in and through sport are not only fundamental components of national and international sport policy but are also crucial factors for good governance, and for maximizing the contribution of sport to sustainable development and peace" [32] (p. 2).

Over the last decade, the number of SFD organizations has continued to grow in tandem with the SFD movement. More programs were initiated in low- and middle-income countries (LMICs) and were backed by international organizations, involving networks of stakeholders outside of LMIC communities. This raised questions about the positioning of international organizations as the "saviors" or "providers" to LMICs [33,34]. Feminist sport researchers argued that the association of transnational non-governmental organizations with major sport brands or 'entanglement of privatization' [35] (p. 522) may reproduce marginalization and inequalities through SFD programs, the initial societal problem that these programs intend to solve in LMICs [36]. Such power dynamics have concerned researchers about the process of translating the effort into the desired developmental outcomes [37–39]. Researchers are urged to consider the systems and environments encompassing these programs, as these external factors could influence the outcome due to gendered socialization in sport [40].

## 2. SFD Theories and Frameworks

SFD is defined as "a social movement that seeks to improve lives through the use of sport and physical activity, and to advance sport and broader social development in disadvantaged communities" [41] (p. 370). In this social movement, sport is used as an intervention in a conventionally complex social development setting, hence it is also perceived as "a social phenomenon observed in the intersection between many disciplines" [42] (p. 1). As a result of the increasing interest in these myriad of disciplines within or related to SFD, there is a burgeoning research area in the field of SFD [2]. The growth was witnessed in both the conceptual and empirical literature, especially in the past decade [43]. Welty Peachey and Hill [43] examined the theoretical advancement of SFD-specific literature through the identification of studies that apply, engage, or mention key SFD theories or frameworks. The researchers discovered 30 articles from 10 prominent journals that published the most SFD-centered articles and the Journal of Sport for Development. Five primary theoretical/conceptual approaches were identified from 30 articles but only one applied a framework to address gender inequality. The theoretical/conceptual approaches

were originally derived from the SFD space and not from other home disciplines such as management studies and feminist theories [44]. These approaches included (a) Sugden's [45] ripple-effect model, (b) Lyras and Welty Peachey's [21] sport-for-development theory (SFDT), (c) Schulenkorf's [46] sport-for-development framework (S4D), (d) Coalter's [18] program theory, and (e) Schulenkorf and Siefken's [47] sport-for-health model. The ripple-effect model conceptualized by Sugden [45] illustrates the impact of change from an intervention as the ripple effect—the center of the intervention (participants) will clearly experience the impact and measurement of the outcomes is easier; as compared to next level or indirect stakeholders (i.e., families, communities, and members in the society), the impact will diminish moving up the level. The SFDT by Lyras and Welty Peachey [21] features five key components: (a) impact assessment, (b) organizational, (c) sport and physical activity, (d) educational, and (e) cultural enrichment. The defining characteristic of this theory lies in the cultural components—the authors believe that this interdisciplinary framework can be exercised to its best outcome when incorporating non-traditional sport management best practices, enhanced through the cultural lens [43]. Schulenkorf's [46] sport-for-development framework focuses on the process and management of SFD projects. This framework does not address a specific focus but it preceded the sport-for-health model, with more flexibility and catering to the programming of SFD projects. Coalter [18] derived the program theory based on the assessment of four sport-based interventions in the United Kingdom. This theory also relied on the understanding of other intervention theories that dissect the relationship between programs and outcomes. This theory suggests that change is most probable through the social relationships between the program leaders and participants. Sports have a pivotal role in social development settings but the relationships are pivotal in the change of values, attitudes, and behaviors. According to Coalter [48], the sport plus model was preferred because sport plus programs facilitate long-term participation and aim for change from within, whereas in plus sport programs, sports are used to attract participants and these programs are usually executed in a short period of time to meet certain goals. The ripple-effect model [45], the SFDT [21], and the program theory [18] will be used to guide the discussion of our findings in this review, focusing on the degrees of outcomes and how the participants' relationships with others were interpreted as change.

*Literature Review*

Since the launch of the SDGs during the United Nations General Assembly in 2015, research has been focusing on the purpose, conception, and politics of these developmental goals [49]. According to Lindsey and Darby [50], the predecessor of the SDGs—the MDGs—were 'tightly focused and relatively discrete' (p. 795). The MDGs were criticized for various shortcomings and one of them was its inadequate alignment with key human rights principles in the context of equality [51]. In contrast, the SDGs are more cross-cutting and universal in terms of their intersectional characteristics and geographical targets. The individual SDGs rely on and influence the development agendas related to other SDGs [52]. The universal SDGs were designed to tackle developmental challenges in both developed and developing countries [53]. Coined as the "important enabler" of the 2030 Agenda for Sustainable Development, SDGs have yet to be academically explored. There is also a paucity of empirical research in determining the relationships between sports and the SDGs [50]. SDG 5 covers a wide scope of targets and indicators [54]; however, none of these targets and indicators are linked to sports or mention sports as a tool to achieve gender equality and women empowerment [29,30]. Nevertheless, sport-based interventions or SFD to tackle the SDGs including SDG 5 have been widely practiced since the conception of the SDGs.

For SFD, the earliest reviews on the literature of SFD can be traced back to 2007 through the work of Kidd and Donelly [55] for the Sport for Development and Peace International Working Group Secretariat, followed by Cronin [56] and van Eekeren et al. [57]. All three reviews shared the similarity of being a commissioned project for organizations

and stakeholders which were keen to explore the efficacy of SFD in producing evidence-based outcomes. Schulenkorf et al. [2] published an extensive review of the SFD literature published from 2000 to 2014, presenting findings on the trends of authorship, geographical contexts, theoretical frameworks, types of sports, level of development, methodological approach, and research outcomes. Other reviews provide a country-specific lens. Langer [58] evaluated the African SFD landscape through a systematic mapping of evidence, and Whitley et al. [19] examined youth-based SFD interventions from six global cities, followed by a similar review focused on the US. There have been a series of reviews on youth-based sport interventions that deployed different methodological or theoretic approaches [59–63]. Each of these reviews helped address the knowledge gaps in SFD and provided potential research or review foci for researchers in different disciplines. Our aim is to extend this focus by focusing on SDG 5 program outcomes in sport contexts.

Beyond academic literature reviews, Svensson and Woods [64] provided a comprehensive overview of the SFD organizations, identifying 955 SFD organizations. The least represented thematic areas were organizations focusing on gender and disability. Hancock et al. [65] provided a global assessment of SFD programs for girls and women based on data collected from four internet databases with 49 out of a total of 376 identified programs focusing on gender equity. The dearth of the reviews on the published literature about programs designed intentionally to contribute to SDG 5 underlines our rationale for this research.

## 3. Methods

This review draws on the PRISMA approach to a structured literature review. Complete details of this approach are articulated below and presented in Figure 1.

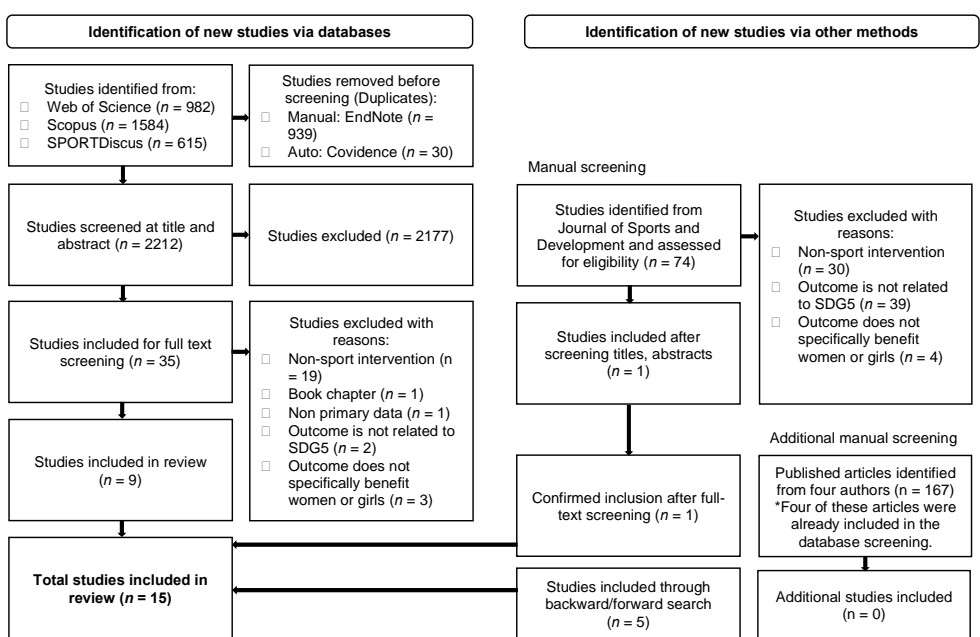

**Figure 1.** PRISMA Flow Diagram of studies screening and selection.

### 3.1. Inclusion and Exclusion Criteria

The inclusion criteria for study selection were: (a) that they were about a program with a major component (50% or more) being sport, which was designed to achieve gender equality and/or women empowerment and (b) that they reported outcomes that benefitted women and girls. The types of outcomes included individual change for women and/or girls and changes that happened to others in the community or society as a whole, such as educational and attitudinal changes with regard to gender. We also included studies that investigated programs targeting men or other genders as long as the intended outcomes

were reported as contributing to gender equality and/or women empowerment. During screening at the abstract and full-text level, whenever facing uncertainties in determining if the program outcomes were within the inclusion criteria, the research team referred to the recommendations from Dudfield and Dingwall-Smith [26] that include four target areas under SDG 5. These outcomes had to have been reported as primary data in these articles.

This review excluded programs that are part of a formal education curriculum or physical education embedded in a compulsory school syllabus. Setting the context as SFD-focused, we ensured that the included studies were research on community-based programs, hence we excluded any sport development programs at the elite level, which includes high-performance collegiate sports. Studies that were not focused on a program, such as those based on grounded theory, were excluded. Programs that encouraged physical activity or exercise were also excluded, as well as a number of studies focusing on non-participants' experiences and outcomes. Only articles published in English were included in this research. This search included peer-reviewed journals and excluded dissertations, reports, conference papers, and commentaries.

### 3.2. Electronic Search

Based on the inclusion criteria, the search terms were constructed and encompassed three domains: (a) sport as an intervention (sport*), (b) population (women OR woman OR girl* OR female), and (c) outcomes that include gender equality or women empowerment (equal* OR empower*). This review included articles from three databases, namely Web of Science, Scopus, and SPORTDiscus, and manual searches of the Journal of Sport for Development. To ensure we did not miss out on any studies central to sport, gender, and development, we searched for scholarly works by four prominent researchers on SFD associated with SDG 5 (Holly Thorpe, Lyndsay M. C. Hayhurst, Megan Chawansky, and Sarah Oxford). One hundred and sixty seven published articles from these authors were discovered on Google Scholar and screened based on the inclusion criteria. Additionally, a backward and forward reference search was also conducted on the articles included after a full-text screening. The database search produced a total of 3181 articles. After removing 964 duplicates, 2212 articles were eligible for screening via the Covidence platform (www.covidence.org, accessed on 28 May 2022).

### 3.3. Screening

Two authors independently screened the titles and abstracts of 2212 articles. Disagreements were resolved by consensus. Thirty five articles were included for a full-text review based on the inclusion and exclusion criteria. From these articles, nine articles were included in the final sample. The manual searches identified another study. From these 10 articles, a backward and forward reference search was carried out. Five articles were subsequently added, resulting in 15 articles for data extraction and analysis. On the manual screening of the four key authors on Google Scholar, there were no additional articles that met the inclusion criteria. The studies by Oxford [66] and Oxford and Spaiij [67] were excluded because of the foci on barriers and constraints of participants' experiences instead of programs and their outcomes. The study by Oxford and McLachlan [68] was also excluded as it investigated a sport program in a mixed-gender setting without the intention to improve gender equity or empower women participants.

### 3.4. Data Extraction and Analysis

A data extraction template was developed to identify: (a) the aim of the study, (b) the outcome of the study, (c) the approach of the study, (d) the instruments or techniques of data collection, (e) the sampling techniques, (f) the number of study participants, (g) the country of the studied program, (h) the month and year when the program started and ended, (i) the type of sports used in the program, (j) the duration of the program, (k) the frequency of the program activities, (l) the profile of the program participants, (m) the number of participants in each cycle of the program, (n) the individual or party who delivered the

intervention, (o) the stakeholders involved in the program, (p) the immediate outputs of the program, and (q) the mid-term and long-term outcomes of the program.

There was a high level of heterogeneity in the extraction, with observed diversity in the interventions (programs), methodologies (study designs), and reported outcomes. Hence, a quantitative meta-analysis and qualitative meta-synthesis could not be conducted. We qualitatively analyzed the 15 articles following the key steps recommended by Braun et al. [69]. As this technique does not require a certain epistemological or theoretical perspective, it is deemed flexible and suitable for analyzing the outcomes [70]. Firstly, we familiarized ourselves with the data and created two primary themes—gender equality and women empowerment—followed by a high-level scanning of the outcomes in the extracted data. Then, we identified the emerging themes in the data based on the guidelines and indicators suggested by Dudfield and Dingwall-Smith (2015) and Kirk (2012). Lastly, we manually mapped the outcomes of these programs accordingly into three categories: the micro-, meso-, and macro-level.

## 4. Results

This section summarizes the 15 included studies and their findings. Table 1 illustrates the studies in this review and their characteristics. Table 2 shows the program information, associated outcomes, and level of impact. The reported outcomes in the articles were first categorized as gender equality and/or women empowerment and then analyzed and labeled as having a micro-, meso-, and macro-level impact.

**Table 1.** The characteristics of Sport for Development studies on programs that contributed to women.

| Authors | Year Published | Study Approach | Instruments/Techniques of Data Collection |
|---|---|---|---|
| McDermott [71] | 2004 | Qualitative | Interviews, participant observations |
| Whittington [72] | 2006 | Qualitative | Interviews, focus group discussions, parent surveys, journal entries and other written documents (secondary data) |
| Van Ingen [73] | 2011 | Qualitative | Interviews, focus group discussions (action research project) |
| Woodcock et al. [74] | 2012 | Quantitative | Questionnaire (cross-sectional survey) |
| Hayhurst [9] | 2013 | Qualitative | Interviews, participant observations, document analysis |
| Musangeya and Muchechetere [75] | 2013 | Qualitative | Interviews, focus group discussions |
| Hayhurst et al. [76] | 2014 | Qualitative | Interviews, participant observations |
| Chawansky and Mitra [77] | 2015 | Qualitative | Interviews, focus group discussions, creative drawing in small groups |
| Hayhurst et al. [78] | 2015 | Qualitative | Interviews, photovoice (participatory action research) |
| Zipp [79] | 2016 | Qualitative | Focus group discussions |
| Meyer and Roche [80] | 2017 | Quantitative | The Attitudes towards Woman Scale for Adolescents, the Gender-Equitable Men (GEM) Scale |
| Bankar et al. [81] | 2018 | Qualitative | Interviews |
| Seal and Sherry [82] | 2018 | Qualitative | Interviews, observations, reflective journaling (participatory action research) |
| Cislaghi et al. [83] | 2020 | Qualitative | Interviews, field observations |
| Lyon et al. [84] | 2020 | Mixed-method | Qualitative tools: journal writing, video diaries Quantitative tools: Single Category Implicit Association Task (SC-IAT), PTSD Checklist for DSM-5, Mental Health Continuum Short Form, Depression, Anxiety, and Stress Scale |

**Table 2.** Program characteristics and outcomes reported in the studies.

| Authors | Country | Profile of Participants | Women Empowerment Outcomes | Gender Equity/Equality Outcomes | Level of Outcome |
|---|---|---|---|---|---|
| McDermott [71] | Canada | White, heterosexual women | - Feeling strong and self-sufficient emotionally and physically<br>- Changed sense of identity | - Null | Micro |
| Whittington [72] | United States of America | Young women who are primarily white, living in rural areas of Maine | - Perseverance, strength, determination, leadership skills, ability to speak out | - Questioned conventional notions of femininity, challenged assumptions of girls' abilities, questioned ideal images of beauty | Micro |
| Van Ingen [73] | Canada | Women and trans survivors of violence | - Healthy aggression in boxing helped redefine women's capacities | - Gender-based violence survivors recognized their voice and body through healthy aggression | Micro |
| Woodcock et al. [74] | Kenya | Young women from different religious backgrounds (aged 10–25 years) | - Female empowerment among participants | - Null | Micro |
| Hayhurst [9] | Uganda | Young women facing pressing inequalities including domestic violence | - Gained confidence, self-esteem, self-respect<br>- Indirectly built financial empowerment and self-reliance | - Ability to fight off sexual advances, refuse sexual relations, and voice their opinions to men | Micro |
| Musangeya and Muchechetere [75] | Zimbabwe | Children and young people (aged 10–24 years) | - Obtained the self-confidence, information, skills, ability, and resolve to make strategic choices to improve their lives. | - Freedom to play sports, understanding of gender differences, positive embodiment<br>- Produced a safe space outside of their homes and families | Micro |
| Hayhurst et al. [76] | Uganda | Young women (aged 10–18 years) | - Improved confidence | - Challenged gender norms | Micro |
| Chawansky and Mitra [77] | India | Young women in urban areas | - Empowerment through sporting activities | - Null | Micro |
| Hayhurst et al. [78] | Canada | Aboriginal young women engaged through a community center | - Improvement in self-determination, increase the sense of anti-colonialism | - Challenged traditional gender roles and stereotypes | Micro |
| Zipp [79] | St. Lucia | Young women who have behavioral problems, suffered from negligence and abuse at home (aged 12–17 years) | - Girls' empowerment, increased self-efficacy | - Learned to challenge gender norms | Micro |
| Meyer and Roche [80] | Senegal | Young men and women | - Results showing female empowerment | - Achieved higher gender equity attitudes<br>- Reduced the perpetuation of destructive gender stereotypes and roles for female youth | Micro |
| Bankar et al. [81] | India | Young women (aged 12–16 years); women mentors (aged 18–24 years) | - Increased ability to think and relate in a collectivized manner<br>- Better negotiation skills and able to support their mentees | - Challenged traditional gender identity<br>- Desexualized public spaces | Meso (mentor) |
| Seal and Sherry [82] | Papua New Guinea | Indigenous women (aged 12–18 years) | - A greater sense of self-efficacy<br>- More conscious about structures of oppression | - Provided spaces to disrupt traditional gendered relations and challenged wider public perceptions | Meso (staff) |
| Cislaghi et al. [83] | India | Young women living in a slum (aged 12–16 years) | - Null | - Parents' acceptance of girls playing sports in public spaces in spite of the patriarchal gender order | Meso (family) |
| Lyon et al. [84] | Australia | Women who suffered child sexual abuse (aged 18–65 years) | - Empowered through participation | - Facilitated the recovery journey from sexual abuse and trauma | Micro |

### 4.1. Study Characteristics

The earliest article was a qualitative study published in Canada [71]. Of the 15 studies, 12 adopted a qualitative approach to investigate gender equality and/or women empowerment SFD programs. This trend echoes the findings of Schulenkorf at al. [2], who reported that empirical SFD articles are primarily qualitative. There were two quantitative [74,80] and one mixed-method [84] studies. Within the 12 qualitative studies, interviews were the primary data collection technique. Three of these studies applied the participatory action research approach where local research participants were informed about the research and empowered to improve their social conditions [85]. Whittington [72], Chawansky and Mitra [77], Meyer and Roche [80], and Cislaghi et al. [83] conducted semi-structured interviews in longitudinal study settings. Chawansky and Mitra [77], Meyer and Roche [80], and Lyon et al. [84] examined pre- and post-intervention measurements of change.

At least one author from every study was affiliated with an academic institution or non-profit organization in North America (United States of America and Canada), United Kingdom, or Australia. These researchers have conducted studies on programs in (a) North America (United States of America, Canada, and St. Lucia), (b) South Asia (India), (c) Africa (Kenya, Zimbabwe, Senegal, and Uganda), and (d) Oceania (Australia and Papua New Guinea). This finding also coincides with Schulenkorf et al. [2]—researchers of SFD programs are predominantly based in Europe, North America, and Oceania but the research on SFD programs has a wider geographical representation. In the present review, some of these studies aimed to investigate the lived experiences of participants and their social relationships with family and peer mentors. These studies reported first-hand experiences as part of the outcomes of the studies; hence, they are included in this review.

### 4.2. Program Characteristics

Although 15 articles were identified for analysis, they represented only 13 SFD programs. Parivartan [81,83] and a martial arts SFD program in Uganda [9,76] were reported in two articles. Football was the most common sport and was used as the intervention tool in four programs. Three programs integrated other sports and/or physical activities in their interventions. Overall, four programs were multi-sport interventions. One of the programs (Parivartan) was designed to use a contact team sport called Kabaddi as a result of consulting with community members, including girl participants, parents, men and women in the community, and the local NGO representatives [83]. Specifically, they believed it was "practical" to engage young girls with this sport as 'it did not require a big field or excessive sport equipment, and it was well known by most children' [83]. The YES Program in Zimbabwe was another program that allowed the participants to choose the sport that they liked from a list of options [75]. The organizer of Because We're Girls was reported to have considered more culturally relevant games for the community [78]. Four out of fifteen programs were set out to mediate the impact of abuse on survivors of violence. Nine programs targeted young or adolescent women and three programs provided mixed-gender activities.

In terms of timing, five articles did not reveal the duration and frequency of the activities. Two of the shortest programs were one-off and lasted as short as 8 days and 23 days, respectively, in the form of outdoor expedition with canoeing as the sport [71,72]. These programs took place before 2010 and intended to create equal opportunities for women to participate in safe and female-only outdoor excursions focusing on canoeing. Hence, these two cases were different compared to the other 11 programs, which targeted a specific women population with social disadvantages. Considering the resources required to deliver the programs, it was unclear how much involvement or influence other stakeholders such as funders and research institutions had over the implementation. Seven programs used the help of local peer leaders or mentors to assist in the execution.

*4.3. Program Outcomes*

As there was high methodological diversity and heterogeneous programs in this review, we aim to describe the outcomes without generalizing what works or what contributes to the effectiveness of interventions. Instead, we underline the importance of context in each program. A study about the effectiveness of a program can only be understood in context [86], and, in the case of this review, the population and the larger societal context in which the studies were conducted. The contribution of these programs to gender equality and/or women empowerment were categorized into the individual (micro-) level, social group or immediate community (meso-) level, and societal (macro-) level, as shown in Table 2. Lyras and Welty Peachey [21] highlighted one of the most substantial building blocks of SFD—assessing three levels of change. According to Burnett and Uys [87], change at the micro-level is always about the 'holistic development of participants' that can be analyzed from their 'personal experiences' and how they perceive 'empowerment' (p. 32). At meso-level, the social networks at the community level (family, groups, and institutions) should be examined to determine if the initial problems that exist within these various networks have been addressed through sport-based interventions. Whereas at the macro-level, the perceived changes can only be analyzed with access to national stakeholders and collaboration to evaluate the long-term impact of using socio-economic and environmental indicators. Table 2 shows the types of program outcomes (gender equality and/or women empowerment) and the levels of each reported outcome. Only two programs, Parivartan and Girls' Empowerment through Sport (GET) Cricket Program [81–83], have achieved more than micro-level outcomes. Ten SFD programs that were investigated in 11 studies met their objectives of contributing to gender equality and women empowerment at the same time [9,72,73,75,76,78–82,84].

## 5. Discussion

*5.1. Reporting of Outcomes through Process and Context*

All studies included in the final review shared the commonality of being a study on a sport-based intervention with desired outcomes or objectives in gender equality and/or women empowerment. We recognize the significance of identifying program outcomes and the level of change, especially for the funders or government agencies behind the program [88], and of informing stakeholders and identifying the transferability to other communities [87,89]. This is especially important and serves as a reminder not to generalize what works when researchers or practitioners consider transferring the same program or study design. Eleven studies recognized structural constraints in the local context where the sport-based intervention took place, unlike the studies based in the United States, Canada, and Australia. While many studies clearly stated the research limitations, Hayhurst et al. [76] has raised concern about the unintended outcomes of sports in marginalized communities such as in Uganda where "cultural inaptness of girls practicing martial arts, may have contributed to the girls' subordination" (p. 157). To address the "Girl Effect" experienced in programs targeting adolescent girls, Hayhurst [9] critically examined the limitations and structural context in preventing individual girls from becoming change agents, using a post-colonial feminist lens. The girls may have experienced personal transformations but the authors also acknowledged that changes beyond the project participants were overshadowed by systemic influence. In these critical examinations, the post-colonial feminist perspective is also pertinent in addressing the girls' agency in a neo-liberal context. In the discourse of localizing SFD programs that are facilitated by global actors, different stakeholders such as international non-profit organizations or even governmental agencies would often refer to macro-level changes whereas grassroot stakeholders or beneficiaries (women and girl participants) were more concerned about the immediate impact, such as the individual behavior change or strengthening of social integration [90]. We argue that a different measure should be applied when communicating with participants and stakeholders to prevent a disconnect between the local aspirations and international organizations' objectives in sustainable development.

*5.2. Research Design*

Most interventions did not extend to the broader community. Bankar et al. [81] engaged with mentors' capacity building as part of a larger intervention and the outcome was assessed post-intervention. Based on Sudgen [45], the impact, similar to the "ripple effect", will decline for the groups that are further away from the center of the intervention (girls or women). Most changes occurred for the participants immediately or in the short term, whereas the meso- and macro-level outcomes were only visible in the medium- to long-term on the basis that the implementation as well as the monitoring and evaluations are sustained [91]. In Cislaghi et al. [83] and Seal and Sherry [82], meso-level outcomes were found when data collection was conducted with family members and staff, respectively. While these indirect stakeholders were not the target participants of the programs, the broader program effects were incorporated into the program and evaluation design. Coalter [18] highlighted the defining success factor of how some programs work when the social relationships between participants and leaders improve. We also argued that a pre-intervention assessment is crucial to define the amount of change that occurred over time or to clarify if such a change could be entirely due to the intervention [21]. Pre- and post-intervention surveys or measurements are critical in assessing the knowledge attainment and attitude change towards gender [92]. While highlighting that the programs in the research are indeed contributing to SDG 5, the researchers are aware about and situated the context of the research in most of the studies included in this review. We advocate for such practice and believe that when considering exporting a program to another country or different context, the local needs and challenges should not be neglected as this could eventually hamper the sustainability and uptake of the program in a new community or social climate [93]. As such, we posit that a robust evaluation scheme and culture-informed indicators should be pre-determined during the design of an SFD solution. Demonstrating the details (e.g., indicators, tools, and approach) opens the door for sport management theorists and practitioners to advance sociological and feminist work in the area of SDG 5.

## 6. Implications for Evaluation and Management of SFD Programs

Evaluation is not merely a post-intervention procedure but a pre-requisite in designing a program or enacting a policy. While Coalter [16] and Houlihan [94] concurred that evaluation or impact assessment informs stakeholders about the success or failure of a solution, the role of evaluation extends further; the evidence on the outcomes and impact facilitates the creation of similar initiatives in the SFD field [95]. Based on Sherry et al. [95], the social context, approach, tools (e.g., the types of sports, and combination of sport and other non-sport activities), and the aim of the program are considerable areas in weighing the success of a program and the feasibility of replication. In circumstances where funders aim to use evaluation data to hold the implementation team accountable, the roles of evaluation are twofold: (1) collecting pre-determined datasets on specific outputs or outcomes and (2) measuring these data against the agreed amount or level of outputs or outcomes [96]. One of the critical concerns in impact measurement owing to neo-liberalism and power imbalance in the SFD network is the subdued voices of the front liners (program delivery teams) and beneficiaries (e.g., disadvantaged women groups) [3,97]. This recognition of problematic evaluation charted a way forward for switching the role of evaluator or researcher to facilitator or collaborator. Through this review, we urge researchers to revisit the roles and power dynamics of researchers with the local community prior to the research. Included in this review, Hayhurst et al.'s [78] approach of post-colonial feminist participatory research used innovative tools such as sharing circles and photovoice to discover contextual constraints among the local community, which were improbable to unearth if the conventional researcher–respondent relationship was in place. The evidence of girls and women living in the nascent and entrenched structural resistance were apparent in Hayhurst [9], Hayhurst et al. [76], and Hayhurst et al. [78]. Recognizing these external barriers imposed on disadvantaged groups, especially on women living in LMICs, Ahmad et al. [98] argued that a feminist monitoring, evaluation, and learning approach can uncover the persisting

imbalance in SFD projects that involve diverse groups of stakeholders. Due to its multidisciplinary nature and complex organizational environment [99], research designs are pivotal to determine how the data are collected, which domain of knowledge is prioritized, and what to do about evidence that reveals the systemic imbalance [98], which hampers the proliferation of the meso- and macro-level impact.

The precedents discussed in the present review also imply that a theoretical or conceptual framework of women-focused SFD programs is warranted due to the needs of (1) locating gender-empowering research methods while producing situated knowledge [100] and (2) incorporating impact assessments within the framework to plan, design, and deliver a holistic program contributing to SDG 5. The existing SFDT brought forward the importance of impact assessment by encompassing it within the theoretical framework derived from promoting peace between community groups [21]. For managing SFD programs that involved higher complexity such as those related to SDG 5, a more innovative and human-centered approach such as design thinking is desirable to approach the tension and complexity due to the number of stakeholders involved [101]. We also question the feasibility of incorporating learning from a decolonized and feminist monitoring and evaluation perspective into sport management to destabilize the top-down mentality among researchers and donors, where academics are assumed as the source of knowledge and donors as the ultimate benefactor.

## 7. Conclusions

Our review aims to further the discussion on the types of studies that are designed to examine women-focused programs intended for empowerment and equality. In the included 15 studies, the evidence of positive outcomes was presented in two major forms: (a) studies about the program outcomes through intervention and (b) studies on how to enhance the outcomes through betterment of the intervention. While all studies reported that micro-outcomes have been achieved, macro-level change has not been investigated. In most of these studies, a common set of indicators that measure sport-based impacts on SDG 5 is absent. We argue that wider implementation as well as impact assessment could be facilitated by the partnership of state and non-state players [102]. The involvement of state and non-state players is critical in configuring indicators relevant to policy making and local-level development [103]. More in-depth and larger-scale studies can help explain the narrative of how sports are improving the lives of women [11,68,104]. A feminist monitoring, evaluation, and learning approach redefines the knowledge creators (the managerial team and participants) and can build the capacity of organizations and the SFD project staff [99]. We recommend participatory research to involve more stakeholders in the planning, design, implementation, and evaluation of the outcomes through longitudinal, multi-study research. The participation of women and girls in the programming and evaluation is pertinent to establish beneficiaries-focused indicators alongside funders-oriented measurements. Consequently, participatory research can tackle the displacement of scope, evangelical beliefs that sport is inherently good [105], and developmental issues such as celebrity humanitarians in LMICs [106]. This setting can include pre-intervention data collection to ensure all parties are informed and empowered to take action throughout the intervention and after the research is completed. Researchers could assist practitioners in developing programs by ensuring clear and measurable goals—"who will make what change, by how much, where and by when" [107] (p. 16). For the past 20 years, many programs were initiated and investigated in Africa, North America, and Oceania. We urge researchers based in Asia to explore SFD and provide more culturally nuanced insights to dissect how gender equality and women empowerment through sport has evolved. We also acknowledge the limitations of the literature search on three databases and the exclusion of books/book chapters and grey literature such as commentaries and conference papers. Lastly, we propose conceptualizing sport management theories or SFD frameworks that are multidimensional and comparable to SFDT with a gender interpretation. We also support sport management studies that acknowledge the stark nature of feminist or sociological

literatures and vice versa. To conclude, we invite scholars to consider how sociological evidence in the feminist, decolonizing paradigm can inform practice in contributing to SDG 5 through sports. By addressing the nexus between theory and practice, we hope to bridge the theory–practice divide in the domains of gender and management.

**Author Contributions:** Conceptualization: Y.-Y.C., E.S., and S.K.; funding acquisition: S.K.; investigation: Y.-Y.C. and S.H.; methodology: Y.-Y.C., E.S., S.H., and S.K.; software: E.S.; supervision: E.S. and S.K.; writing—original draft: Y.-Y.C.; writing—review and editing: E.S. and S.K. All authors have read and agreed to the published version of the manuscript.

**Funding:** This research and APC was funded by the Universiti Malaya Research Grant (RP047A-17HTM).

**Institutional Review Board Statement:** Not applicable.

**Informed Consent Statement:** Not applicable.

**Data Availability Statement:** Not applicable.

**Conflicts of Interest:** The authors declare no conflict of interest.

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
