# Peer review of "Sport for Development Programs Contributing to Sustainable Development Goal 5: A Review"

_sustainability, doi:10.3390/su14116828_

Round 1
Reviewer 1 Report
-
Author Response
Thank you for your time and review.
Reviewer 2 Report
Its important to have a section in the lit review which actually details SDG5. You also need to define empowerment.
'Girl Effect', expand upon this in the conclusion as this is quite central given that sports companies are using women's bodies and sexuality, appropriated and commodified to sell good. Who then is benefiting from empowerment? You do mention p13 and onwards. Do cite Dan Brockington who has written about celebrity status enhancing poverty reduction in Uganda for example. Of the studies we do not get the sense they were actually writing about development and sustainbility, this conclusion has been drawn. But who are the leaders trying to include women, I'm not sure that came across. If its corporate USA the criticisms need more nuance
But, in a nutshell you have done a meta analysis of getting women into sport, and summarised this, and have drawn SFD conclusions, when many sfd were not realized. You need to link value of study to the Global South.
There are no UN Women references (this is a sustainability journal) and few sustainability references, for a paper that seems more suited to many on the journal list. In conclusion be clear on the sdg5.
Reviewer 3 Report
The discussed topic is important and interesting in both the scientific and the pragmatic scope. The article is of a review character, but also takes up ordering and postulatory themes, bringing new content to science. The structure of the article is correct, as are the presented methodological issues. A satisfactory range of scientific literature has been used. The presented content can become the basis for further research, both in theoretical-review and empirical aspects. In addition, they can inspire public sport management bodies that take into account sustainability criteria.
Author Response
Thank you for your positive comments.